# Global mean sea level likely higher than present during the holocene

Roger C. Creel [1,2] ✉, Jacqueline Austermann [2], Robert E. Kopp [3], Nicole S. Khan [4], Torsten Albrecht [5,6] & Jonathan Kingslake[2]

Global mean sea-level (GMSL) change can shed light on how the Earth system responds to warming. Glaciological evidence indicates that Earth's ice sheets retreated inland of early industrial (1850 CE) extents during the Holocene (11.7-0 ka), yet previous work suggests that Holocene GMSL never surpassed early industrial levels. We merge sea-level data with a glacial isostatic adjustment model ensemble and reconstructions of postglacial thermosteric sea-level and mountain glacier evolution to estimate Holocene GMSL and ice volume. We show it is likely (probability $P = 0.75$) GMSL exceeded early industrial levels after 7.5ka, reaching 0.24 m (−3.3 to 1.0 m, 90% credible interval) above present by 3.2ka; Antarctica was likely ($P = 0.78$) smaller than present after 7ka; GMSL rise by 2150 will very likely ($P = 0.9$) be the fastest in the last 5000 years; and by 2060, GMSL will as likely than not ($P = 0.5$) be the highest in 115,000 years.

The time interval extending from the start of the Holocene interglacial period (11.7 thousand years ago, ka) to the start of the industrial era (1850 CE, hereafter 'early industrial') marked the final melting of the two largest Northern Hemisphere ice sheets and the onset of a warm, stable interglacial. During this interval, polar temperatures may have temporarily exceeded early industrial temperatures by several degrees[1,2]. Studying global mean sea level (GMSL) during the Holocene, therefore, offers perspective on ice-sheet sensitivity to past and future warming.

Previous reconstructions of Holocene GMSL are mostly based on local relative sea level observations. Relative sea level (RSL) deviates from GMSL in part due to glacial isostatic adjustment (GIA), which describes the gravitational, rotational, and viscoelastic deformational effects of water and ice loading on a gravitationally self-consistent solid Earth[3]. During the Holocene, these effects cause RSL in areas below former ice sheets (e.g. in Canada and Northern Europe) to fall due to isostatic rebound and rise at the edges of these ice sheets due to peripheral bulge subsidence, while far from ice sheets, GIA causes an array of smaller-order RSL changes due to far-field effects including ocean siphoning and continental levering[4]. These far-field effects can

explain much of the highstand observed at low-latitude sites during the mid-late Holocene[5]. GMSL studies typically use GIA modeling to jointly refine ice-sheet reconstructions and solid Earth structure until the predicted RSL estimates fit observational constraints, then calculate GMSL from the reconstructed ice volumes[6-9]. For example, Peltier and colleagues iteratively modified a post-glacial ice reconstruction to fit geodetic uplift rates and RSL observations at a small set of far-field sites and found that GMSL was less than a meter below present levels at 6 ka and gradually increased to reach within 5 cm of present levels by 2 ka (Supplementary Fig. 1, ref. [9]). Lambeck and colleagues, on the other hand, iteratively inverted far-field RSL observations for mantle viscosity and continental ice distributions to find that GMSL was more than $3 \pm 0.7$ m below present at 6 ka and remained below present throughout the Holocene[6], a finding supported in a similar study by Bradley et al.[8]. None of these studies conclude that the Antarctic Ice Sheet may have been substantially (> 0.2 m GMSL equivalent) smaller earlier in the Holocene than in the early industrial period.

In contrast to the models mentioned above, near-field evidence suggests that several sectors of the Antarctic Ice Sheet retreated inland of their early industrial grounding lines before re-advancing during the

[1]Department of Physical Oceanography, Woods Hole Oceanographic Institution, Woods Hole, MA, USA. [2]Lamont-Doherty Earth Observatory, Columbia University, New York, NY, USA. [3]Department of Earth and Planetary Sciences and Rutgers Climate and Energy Institute, Rutgers University, New Brunswick, NJ, USA. [4]Department of Earth Science and Swire Institute of Marine Science, University of Hong Kong, Hong Kong, Hong Kong. [5]Potsdam Institute for Climate Impact Research (PIK), Member of the Leibniz Association, Potsdam, Germany. [6]Department of Integrative Earth System Science, Max Planck Institute of Geoanthropology, Jena, Germany. ✉ e-mail: roger.creel@whoi.edu

mid-to-late Holocene[10–12]. This evidence includes sediment cores from ice-marginal lakes, sea-level indicators from raised beaches, radar observations of englacial structures, geodetic measurements of bedrock subsidence, and radiocarbon dates on sub-glacial organic carbon[13,14]. These polar constraints are supported by regional physics-based ice reconstructions that, with a range of parameterizations, reproduce Holocene readvance[11,15–19]. However, the field evidence and ice-sheet models do not uniquely constrain the timing and amount of retreat and readvance.

There are several reasons why previous sea-level-based studies could have mis-estimated Holocene Antarctic ice volume and GMSL. First, Holocene GMSL variation is expected to be much smaller than the LGM-to-present change, which is the main focus of most of the studies that produced these estimates (though not of ref. 8). Second, these studies only had access to a fraction of the sea-level data now available. Third, the studies may have discounted excess Antarctic melt because of lack of evidence: their publication dates preceded the emergence of recent observational evidence supporting a smaller-than-present Holocene Antarctic ice sheet, and ref. 6 acknowledge that lack of observational evidence precludes independently constraining cryospheric fluctuations between 7 ka and the early industrial. Fourth, they do not account for laterally varying Earth structure in GIA modeling, an omission that could introduce inaccuracies[20]. Fifth, they omit the feedback between ice sheets and GIA, which could cause differences in ice dynamics, especially in West Antarctica, that would in turn affect ice volume and GMSL inference. They also did not include thermomechanical ice-sheet models applied to oceanic/atmospheric forcing. Sixth, they do not include thermosteric sea-level change or mountain glacier fluctuations. Seventh, they underestimate uncertainties by preferring single best estimates of GMSL or narrow confidence intervals. Lastly, they may have overlooked the possibility of GMSL higher than present because of algorithmic design that precludes ice-sheet histories with Holocene ice volume smaller than at early industrial[8], or that enforces a single optimal history even when a second best-fitting curve exists that includes GMSL higher than present[6] (see Supplementary Fig. 1). Lack of agreement in Holocene GMSL predictions led the International Panel on Climate Change Sixth Assessment Report (IPCC AR6) to assess, with medium confidence, a mid-Holocene (6 ka) GMSL 90% confidence interval ranging from 3.5 m below present-day to 0.5 m above present-day, the spread of which is chiefly explained by the uncertain history of the Antarctic Ice Sheet during the Holocene[21].

To improve our understanding of Holocene GMSL and provide a far-field constraint on Holocene Antarctic Ice Sheet change, we pair a postglacial (23 ka to 1850 CE) database of RSL observations with an ice-sheet ensemble via an algorithm that approximates the influence of laterally varying Earth structure without employing 3D GIA models (Supplementary Fig. 2, see "Methods" for details). The database includes 10,253 sea-level data (Fig. 1) from low- to mid-latitude geological and biological archives such as salt marshes, mangrove swamps, coral reefs, and deltaic sediments[22]. The sea-level model 'prior' consists of a range of RSL predictions from an ice-sheet ensemble that combines several Northern Hemisphere simulations and 278 Antarctic simulations from the Parallel Ice Sheet Model (PISM), which span a mid-Holocene (6 ka) GMSL-equivalent range of ~ − 16 to +2 m (Supplementary Fig. 3A)[15,23,24]. We include a large range of Antarctic histories in the ice-sheet ensemble because Holocene Antarctic variability is more uncertain than Greenland Ice Sheet behavior (Supplementary Fig. 4A)[13,25]. The model prior also includes probabilistic estimates of Holocene mountain glacier volume and global-mean thermosteric sea-level change. We use sea-level data and near-field observational constraints from around the Antarctic Ice Sheet to calculate a posterior distribution of GMSL and Antarctic ice change. Because RSL is modeled using GIA, it ensures that we account for processes such as ocean siphoning, which can cause RSL highstands in tropical regions even in the absence of a GMSL highstand[5].

Our approach includes the ~ 10- to 20-fold increase in far-field RSL data over previous studies; the inclusion of mountain glacier and thermosteric sea-level contributions; the large number of ice histories (n = 26,688) and GIA models that we employ (n = 2,135,040); the inclusion of marine- and terrestrial-limiting data, which are often excluded from GMSL studies; and the method that we develop to approximate the influence of laterally-varying solid Earth structure via the region-by-region comparison of an ensemble of 1D GIA models to RSL data. Our algorithm's efficacy is demonstrated with synthetic tests (see "Methods" and Supplementary Fig. 5), which show that our modeling is able, within uncertainties, to reproduce a range of plausible GMSL scenarios. We further demonstrate the fidelity of our approach to local RSL data by producing posterior RSL curves for a range of locations (see Supplementary Figs. 13–16). In addition to inferring Holocene GMSL and Antarctic ice volumes, our approach also allows us to compare the amplitude and rate of Holocene GMSL and Antarctic mass change to projected 21st century sea-level rise and Antarctic mass loss.

## Results
### Holocene global sea level trends
The median of the final Holocene GMSL curve (hereafter the 'median posterior') has three phases: rapid early-Holocene rise, slower mid-Holocene rise, then gradual late-Holocene fall (Fig. 2B). Rates of GMSL rise start to slow after 8 ka−a trend corresponding to the final Laurentide Ice Sheet termination (Supplementary Fig. 6A)[26]. The median

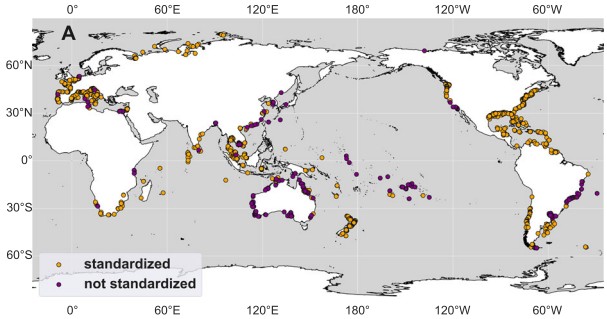

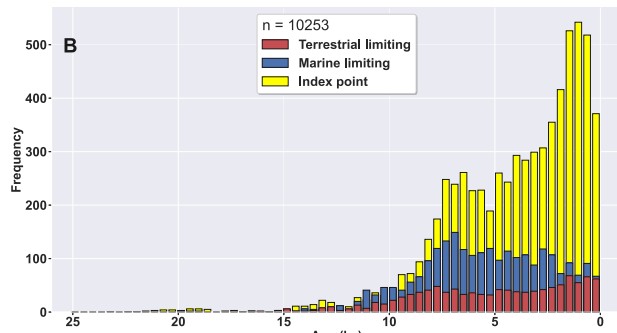

**Fig. 1 | Postglacial relative sea level (RSL) data. A** Geographic distribution of RSL data. Orange markers denote data standardized following procedures agreed upon by the sea-level community (Table S1)[22]; purple markers denote additional data presented as originally published (Table S2). **B** Temporal frequency of RSL data.

Red, yellow, and blue bars in (**B**) indicate, respectively, the number of terrestrial limiting data, index points, and marine limiting data. Bars plot on top of each other. Note that data below former ice sheets are not used in this analysis.

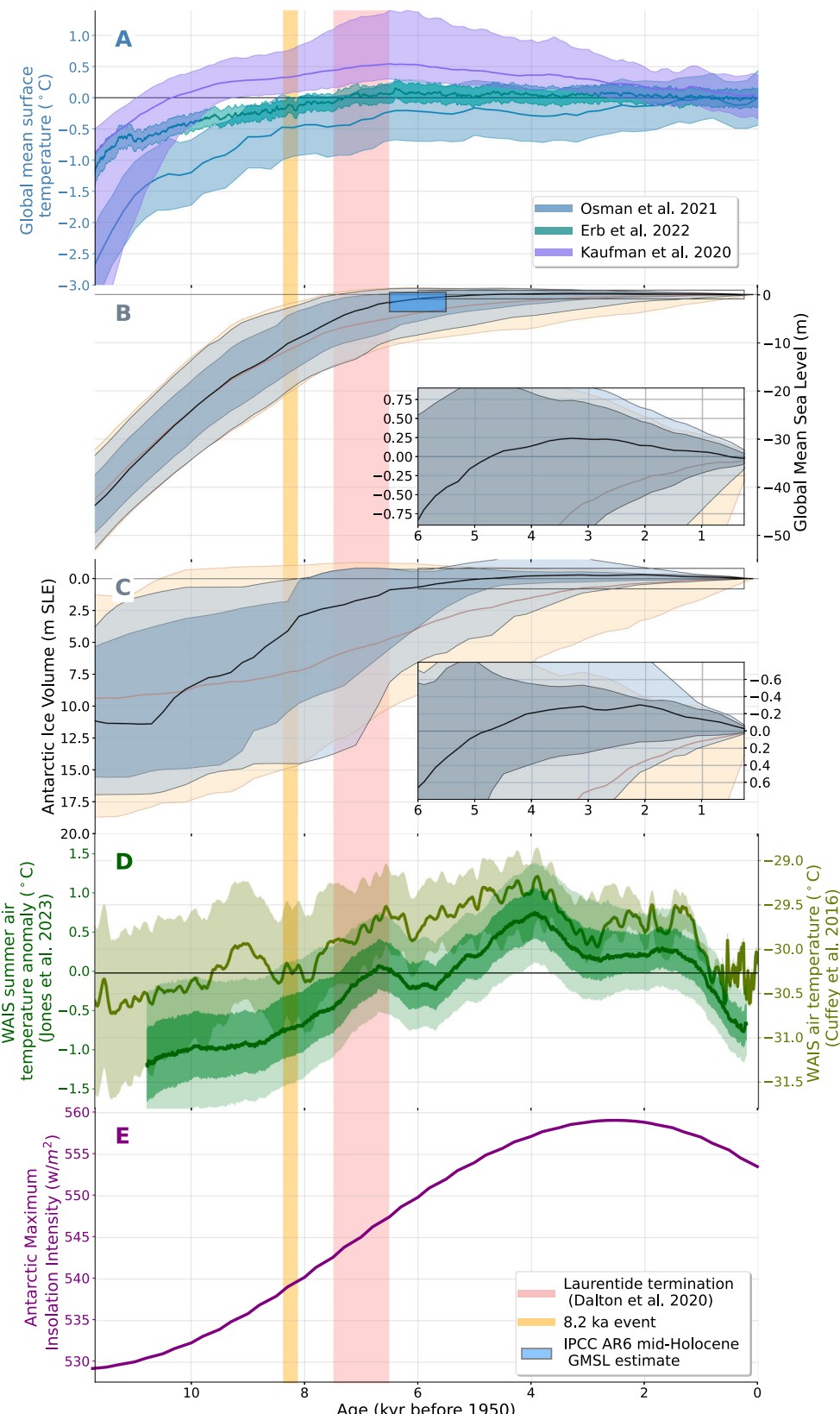

posterior reaches −1 m (−8.6 to 1.0 m, 90% credible interval) at 6 ka, which encompasses the IPCC-AR6 mid-Holocene GMSL estimate of −3.5 to 0.5 m. The posterior 90% credible interval envelopes GMSL estimates from the ANU[6], ICE-6G[9], PaleoMIST[7], and Bradley[8] ice models. PaleoMIST, Bradley, and ANU, which by 6 ka reach −6.6 m, −6 m, −2.9 ± 0.7 m, respectively, fall below the median posterior; ICE-6G, at

−0.4 m by 6 ka, reaches above the median posterior. The median posterior peaks at 0.27 m (−3.1 to 1.0 m) at 3 ka (Fig. 2B inset).

Separate from analysis of the median posterior curve, we also examine the likelihood and amplitude of peak Holocene GMSL (see "Methods" for a detailed explanation of how this is calculated). It is likely (*P* = 0.75) that peak GMSL exceeded 0 m after 7 ka. High

**Fig. 2 | Holocene global mean sea level (GMSL) and Antarctic ice volume compared to climate variables. A** Global mean surface temperature reconstructions[1,37,38]. **B** GMSL. Brown and black lines denote the prior and posterior 50th quantile; tan and light gray bands the prior and posterior 90% credible intervals; and darker gray band the posterior 66% credible interval. Blue box demarcates the Intergovernmental Panel on Climate Change's 6th Assessment Report (IPCC AR6) mid-Holocene GMSL estimate. **C** Antarctic ice volume. Inset boxes in (**B**) and (**C**) have same axes labels as (**B**) and (**C**). **D** Annual air temperature[2]

(olive green, less smooth) and summer air temperature anomaly[29] (forest green, smoother) from the West Antarctic Ice Sheet Divide core. Summer air temperature is 1000 year moving average. Light olive and forest green envelopes in (**D**) are 95% confidence interval; dark forest green envelope is the 68% confidence intervals. Black reference line denotes temperature mean over the last millennium (ref. [2]) and the anomaly relative to that mean (ref. [29]). **E** Maximum Antarctic summer insolation intensity[97,98]. Pink and orange vertical lines indicate final Laurentide termination[26] and the 8.2 ka event, respectively.

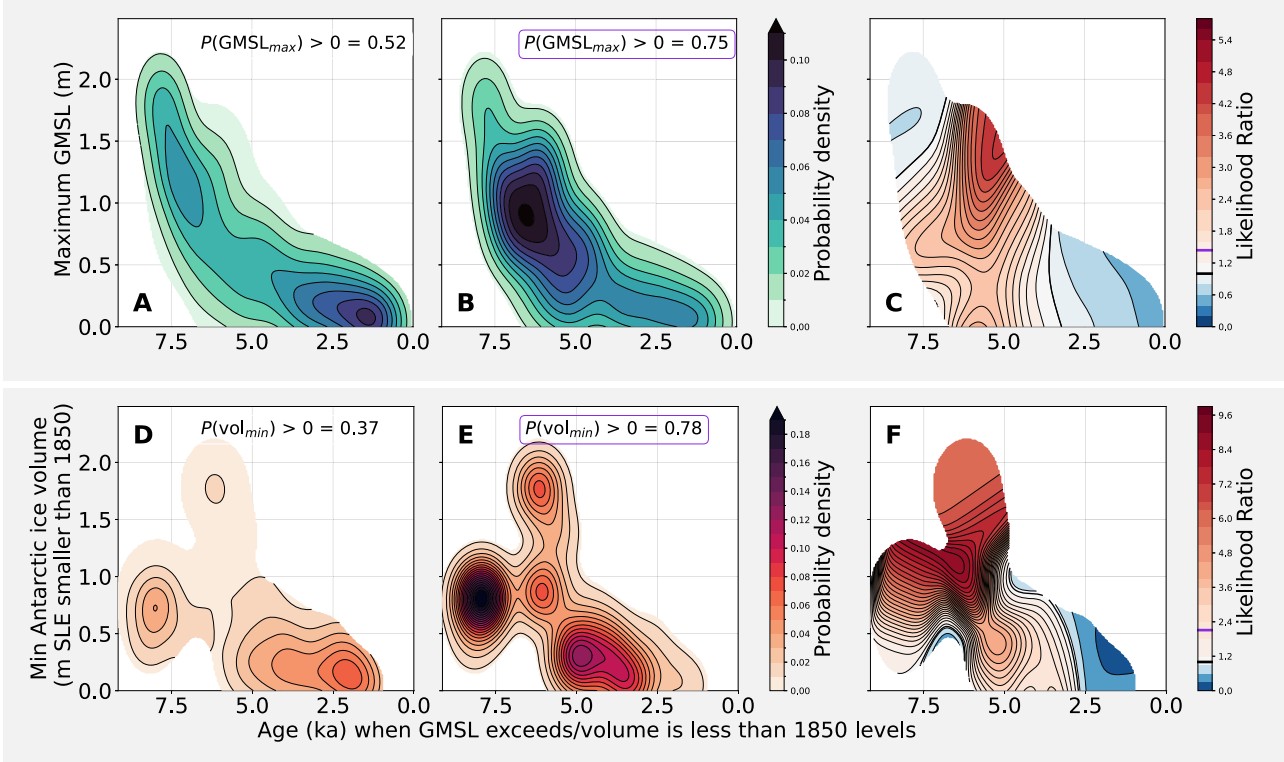

**Fig. 3 | Maximum amplitude and time of pre-industrial exceedance of global mean sea level (GMSL) and Antarctic ice volume minimum for global ice-sheet scenarios.** Top row (**A**): prior probability distribution of model maxima and time each model first exceeds present levels, i.e. distribution without weighting by relative sea level (RSL) observations and Antarctic constraints. **B** Posterior distribution of (**A**). $P(GMSL_{max})$ denotes that probability that prior (**A**) or posterior (**B**) GMSL exceeded present levels. **C** Likelihood ratio, calculated as the ratio of (**B**) to (**A**), which represents the degree to which the data constraints have increased the likelihood of a given maximum GMSL. Bottom row (**D/E/F**): Prior distribution,

posterior distribution, and likelihood ratio for Antarctic ice volumes. $P(vol)_{min}$ denotes that probability that prior (**C**) or posterior (**D**) Antarctic Ice Sheet volume was smaller than at present. Black line on colorbars for (**C**) and (**F**) denotes a likelihood ratio of 1, which indicates no increase in likelihood; Purple line on colorbars denotes the likelihood ratio of the probability that GMSL exceeded present levels or the Antarctic Ice Sheet was smaller than today; values higher than one indicate that exceedence or smaller-than-present volume are more likely in the posterior than the prior. Ice volumes are shown in GMSL equivalent units.

probability density in posterior peak GMSL around 1 m around 6 ka (Fig. 3B, C) indicates that data constraints upweight the subset of models that peak at this GMSL and time. However, when the whole ensemble is considered, there are many models that do not peak at this time and are also likely, which leads the median posterior at 6 ka to be lower than 1 m (Fig. 2). While our results favor GMSL overshooting pre-industrial levels by up to 1 m in the mid-late Holocene, they also leave open the possibility ($P = 0.25$) that Holocene GMSL did not exceed pre-industrial levels.

Our analysis reveals details of Antarctic ice volume that agree with recent field evidence but differ from previous GMSL studies. We find that the Antarctic Ice Sheet likely ($P = 0.78$) shrank below its 1850 volume during the Holocene. The Antarctic Ice Sheet was likely smaller than present after 3.9 ka (0.2–7.9 ka, 90% credible interval) and reached a minimum of 0.3 m ( − 0.5 to 0.9 m, 90% credible interval) GMSL equivalent smaller than pre-industrial at 2 ka (Fig. 2C and inset).

This timing aligns with geomorphological, sedimentary, and geophysical evidence from Antarctica[13]. We find that Antarctic ice volume closely tracked both insolation and West Antarctic terrestrial temperature (Fig. 2C, D, E). The median posterior Antarctic ice volume estimates are smaller than the prior median Antarctic distribution for virtually the entire Holocene, during some intervals by up to 4 m GMSL equivalent (Fig. 2C). Further, evidence from sea-level data and near-field constraints heavily favors an Antarctic Ice Sheet that shrinks to ~0.5–1 m smaller than its present volume between 8 and 4 ka (Fig. 3D–F)–a result that aligns with the favored minimum mid-Holocene Antarctic volume range (−0.4 m to  −1.0 m) of ref. [16]. Because differences between posterior and prior distributions indicate that the sea-level data constraints have added information to the model, this result demonstrates that intermediate- to far-field RSL data can help distinguish detailed variations in Holocene Antarctic ice volume.

We also note that the distribution of peak Antarctic ice melt, which has high probability density starting at ~8 ka, is related to but distinct from peak GMSL, which has highest probability density starting 6 ka (Fig. 3B, E). This difference happens because Greenland Ice Sheet's melt peaks occurs later in the Holocene starting around ~7 ka (Supplementary Fig. 4) as does a small amount of excess Holocene melt from mountain glaciers starting at ~6 ka (Supplementary Fig. 8) and a modest GMSL contribution from thermal effects as early as ~11 ka or from ~6 to 4 ka (Supplementary Fig. 7). These diverse sources of GMSL rise mean that high GMSL in a model can be caused by a small Antarctic Ice Sheet, a small GRIS, small mountain glaciers, large thermal expansion, or some combination of the four sources. On the other hand, excess melt from Antarctic Ice Sheet can (at any point) be masked by a larger Greenland ice sheet or colder ocean temperatures.

Postglacial RSL data and near-field Antarctic constraints are not able in this modeling framework to differentiate between the other sea-level contributors, including Northern Hemisphere ice sheets, mountain glacier histories, and thermosteric effects; the posterior distributions of these contributors therefore do not change relative to the prior. This is likely because of the small amount (<0.2 m) that thermosteric effects (Supplementary Fig. 7) and mountain glacier histories (Supplementary Fig. 8A) likely contributed in the last 6000 years as well as the smaller number of Northern Hemisphere ice-sheet simulations included in our model relative to the number of Antarctic simulations.

Future efforts to compile high-quality RSL databases should focus on observations close to the Antarctic and Greenland ice sheets. While near-field data were excluded from this study because of their sensitivity to Earth structure, they are regularly included in construction and/or validation of ice histories (cf. ref. [9]) and would therefore be a valuable contribution to future work to reconstruct the history of the Antarctic Ice Sheet. Times and areas with low data coverage in the far-field—e.g. data between 8 and 12 ka and from Africa, India, Siberia, the Pacific, and China—could also help to distinguish among differing Antarctic ice histories. The need for more data is underscored by sensitivity tests in which we re-ran our analysis using only index points, only index points compiled according to agreed-upon community standards[22], and only standardized data including both index points and limiting points (Supplementary Fig. 11). We find that median GMSL with all data, when compared to GMSL estimated with data subsets, is up to 0.6 m higher between 12 and 9 ka, 2–5 cm higher at 6 ka, and less than 1 cm lower at 3 ka. Median Antarctic ice sheet volume with all data is up to 0.5 m higher in the early Holocene than median Antarctic ice sheet volume calculated with data subsets and differs by up to 5 cm in the last 6000 years—differences that are smaller than our model uncertainties. The consistent pattern in these data—the inclusion of more data leads to higher GMSL—suggests that more data could have a meaningful impact on our estimation of GMSL, which motivates future work to collect, aggregate, and standardize Holocene sea-level data.

## Discussion

### Antarctic Ice Sheet driven by local temperature

Recent debate surrounding future Antarctic Ice Sheet instability has focused attention on the processes responsible for Antarctic Ice Sheet behavior during the Holocene. Antarctic ice volume may have followed polar temperature[17], as likely happened in Greenland[27]. Alternatively, Antarctic readvance may have been driven by GIA, because isostatic rebound in areas of ice-sheet retreat can reground ice sheets[10,11,28]. While our model does not provide causal evidence to distinguish between these hypotheses, we find a significant cross-correlation (-0.4–0.6) between Antarctic ice volumes and local temperature records based on isotopic evidence from the West Antarctic Divide ice core[2]. Late-Holocene Antarctic volume lags West Antarctic annual air temperature by 250–650 years across a range of frequencies; a similar (250–500 yrs) lag is present for summer air temperatures (Fig. 2D,

Supplementary Fig. 10). Late Holocene Antarctic ice volume also broadly aligns with the local maximum in summer insolation intensity at 3-2 ka, which likely controls summer temperature[29]. The connection between temperature and ice volume is further bolstered by terrestrial sedimentary records that support a climatic optimum between 6 and 3 ka[30]. A similar relationship is present with marine temperatures. High latitude southern hemisphere marine temperature records from diatom abundances generally infer mid-Holocene warmth followed by late Holocene cooling[31–33], as do aggregated southern hemisphere high-latitude proxies, which indicate that polar waters reached their warmest at ~6 ka[1]. The PISM Antarctic Ice Sheet models in our ensemble are forced by West Antarctic Ice Sheet Divide surface air temperatures, which are generally higher in the time period before 3 ka. Nevertheless, our GMSL (and by extension, Antarctic ice mass) prior is designed to not give a high likelihood to models with excess ice mass in times older than 3 ka (uniform likelihood of GMSL being between −10 m and +2 m at 6 ka, see "Methods"). As a result, there is no major peak in the prior probability density of the GMSL and Antarctic ice mass exceedance before 3 ka (Fig. 3A, D). This changes once data are incorporated, which leads to considerable probability density on GMSL higher and Antarctic Ice Sheet smaller than present levels before 3 ka and as early as 7·5 ka in the posterior estimate (Fig. 3C, E). The correspondence between West Antarctic Ice Sheet temperature and Antarctic Ice Sheet volume points to temperature forcing, both terrestrial and marine, as likely driving mechanisms (Fig. 2C, D, E), and lends credence to arguments that summer insolation, local temperatures, and Antarctic Ice Sheet variations are tightly coupled[34,35]. These links do not preclude other explanations for Antarctic readvance such as isostatic uplift, but rather motivate further work to understand the timing of GIA-driven rebound and its potential role in Holocene Antarctic ice dynamics.

### Perspective on interglacial temperature

Our findings suggest that GMSL and global temperature are decoupled during the Holocene. Estimates of Holocene global mean temperatures, generated from diverse combinations of sea surface temperature proxies, terrestrial temperature data, and climate model outputs, vary from monotonic temperature increase[36,37] to a mid-Holocene temperature peak of between 0.1 ℃[38] and more than 0.4 ℃[1,39] (Fig. 2A). These temperature histories differ from our GMSL reconstruction, which most likely exceeded current GMSL in the mid-Holocene but only reached its maximum in the late Holocene (Fig. 2B). While it is expected that GMSL would lag temperatures, it is important to consider that global mean temperature integrates insolation variation across all latitudes, while GMSL is driven principally by polar ice mass changes, which can lag decades (mountain glaciers[40]), centuries (Greenland Ice Sheet[41]), or millennia (Laurentide Ice Sheet[42]) behind high-latitude temperatures. Efforts to understand Earth's GMSL commitment for each degree of warming regularly use past periods when global mean surface temperature and GMSL were higher than today as analogs for a future warming world[43,44]. Our results indicate that this approach could be improved by instead targeting high-latitude temperature records that characterize the behavior of individual ice sheets. This distinction is particularly important when high-latitude temperatures are out of sync between the northern and southern hemisphere, as likely occurred during the Last Interglacial[45].

### Contextualizing modern sea level rise

A central role of paleoclimate research is to place anthropogenic climate change in the context of natural climate variability. Here, we do this by comparing our peak Holocene GMSL estimates to future sea-level projections from the International Panel on Climate Change's Sixth Assessment Report[21]. The rate of GMSL rise between 1850 and 2005 was likely ($P > 0.75$) higher than rates over the last 4000 years but unlikely ($P = 0.13$) higher than over the last 7000 years. Rates of future

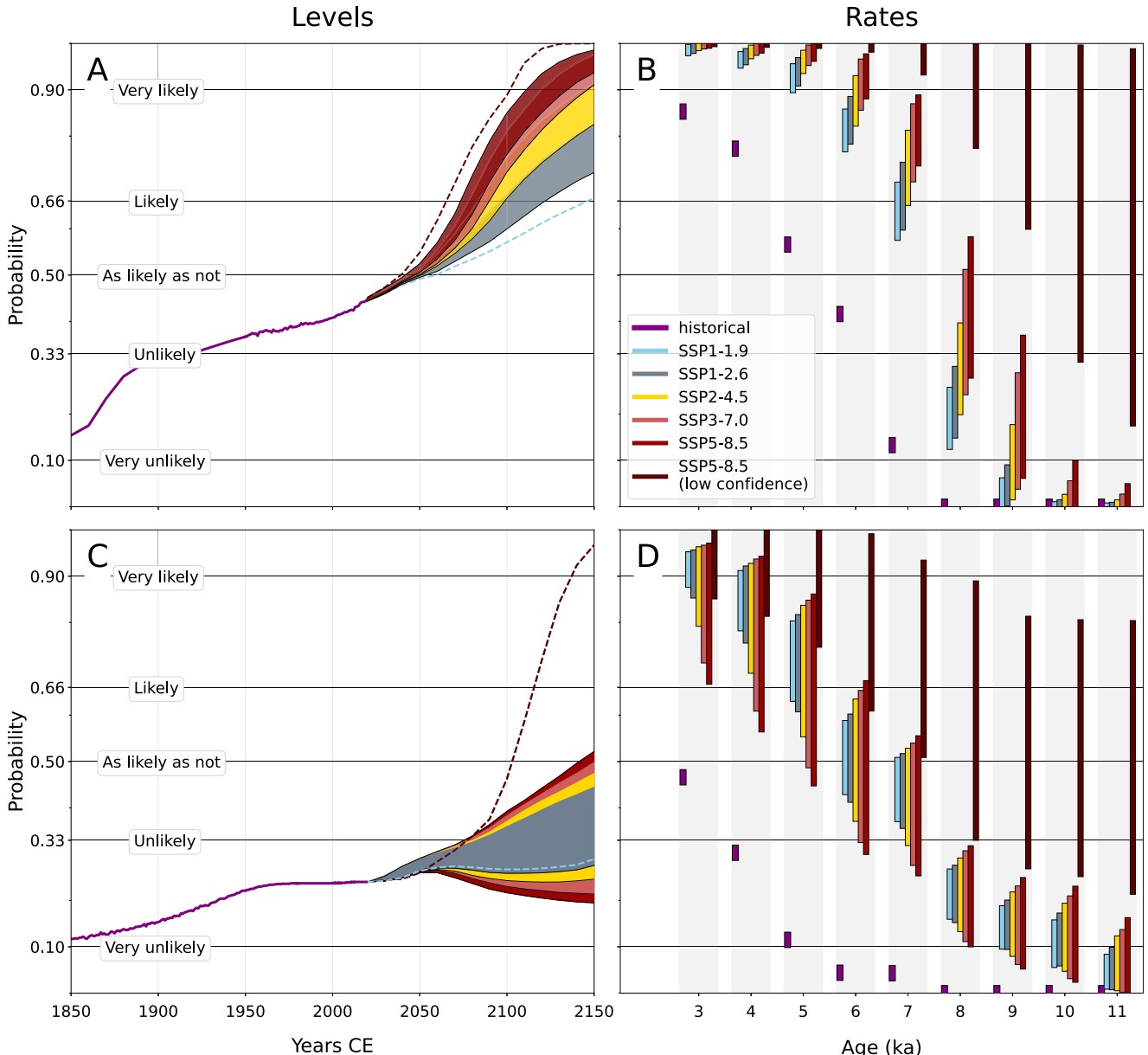

**Fig. 4 | Probability over the period between 1850 and 2150 that global mean sea level (GMSL) and Antarctic Ice Sheet volume change and rates of change exceed Holocene levels and rates of change.** Probability that the level of GMSL (**A**) or Antarctic ice volume (**C**) exceeds the maximum Holocene (11.7 ka to 1850 CE) level. Probability that the rate of historic (1850–2005) or future (2005–2150) change of GMSL (**B**) or Antarctic ice volume (**D**) is greater than the maximum rate of change over the last 3, 4, 5, 6, 7, 8, 9, 10, or 11 kyr. Green lines between 1850 to 1950 (**A**) or 1979 (**B**) represent probabilities from calculated relative to GMSL values from (ref. 93) and (ref. 94, see "Methods"), respectively. Green lines between 1950 (**A**) or 1980 (**B**) and 2020 represent exceedence probabilities calculated relative to observed GMSL or Antarctic ice volume as reported by the Intergovernmental Panel on Climate Change's 6th Assessment Report (IPCC AR6)[21]. Solid color bands in (**A/C**) represent future exceedance probabilities from 2020 to 2150 calculated relative to likely ranges for selective socioeconomic pathways (SSP) 1–2.6 through 5–8.5 for processes in which there is at least medium confidence, as assessed by IPCC AR6. Dashed sky blue and dark red lines in (**A/C**), respectively represent the lower end of the likely range for SSP1-1.9 and the upper 83rd percentile of low-confidence projections for SSP5-8.5. Green bars in (**B/D**) represent the probabilities that the average rate of historical sea level rise (1850–2005) exceeds the maximum rate of Holocene sea level rise during the last 3–8 kyr, as noted in vertical gray bars.

GMSL change between 2005 and 2150 will very likely (*P* > 0.9) be the highest in the last 5000 years and more likely than not (*P* > 0.5) the highest since the Laurentide Ice Sheet collapsed at around 7 ka (Fig. 4B). Future GMSL will more likely than not (*P* > 0.5) exceed maximum Holocene GMSL by 2060 under all emissions scenarios (Fig. 4A). By 2150, future GMSL will likely (*P* > 0.66) be higher than peak Holocene GMSL under low emissions (SSP1-2.6) and very likely (*P* > 0.9) higher under high emissions (SSP5-8.5) (see "Methods").

By contrast, rates of Antarctic Ice Sheet volume loss between 1850 and 2005 were unlikely (*P* = 0.12) to have been higher than earlier in the past 4000 years. In the future, it is more likely than not (*P* > 0.5) that

rates of Antarctic Ice Sheet shrinkage will be higher than in the last 4000 years, but unlikely (*P* < 0.33) that they will be higher than during the last 8000 years (Fig. 4D). However, when incorporating the effects of poorly understood 'low confidence' ice-sheet processes, rates of GMSL rise under the highest emissions scenario (SSP5-8.5) may exceed any Holocene rates (Fig. 4B). And because of the lag between temperature and ice-sheet mass loss, should high emissions continue beyond the 21st century, GMSL would likely (*P* > 0.66) continue to rise faster than any Holocene rates for several hundred years, only slowing after the complete collapse of the West Antarctic Ice Sheet[46]. Our results therefore add urgency to the need for a better understanding of

Antarctic Ice Sheet dynamics during the present interglacial, the mechanisms that drive these dynamics, and the implications for future Antarctic ice stability.

Earth's climate in the past 9000 years has been unusually stable relative to past environmental changes. This 'safe operating space'[47] enabled the rise of agriculture, civilization, and industrialization. Our results indicate that projected future rates of GMSL rise exceed rates for the past 7000 years but are comparable to those that early Holocene civilizations experienced. However, this equivalence belies the vast differences between how modern and ancient human societies adapted to sea-level rise. Humankind prior to 8 ka consisted of fewer than 50 million people, many of whom were migratory[48]. Modern human civilization in the 21st century is projected to near 10 billion people[49], hundreds of millions of whom live in permanent coastal communities that cannot be relocated inland. The 'safe operating space' for sea-level rise will be smaller for future generations than it was for past cultures.

## Methods

### Sea level data

Sea-level observations are taken from two sources: HOLSEA-standardized papers ($n = 7923$, Table S1), hereafter called HOLSEA data[22], and published sources not yet compiled into HOLSEA format, hereafter called non-HOLSEA data ($n = 2330$, Table S2). To be included, non-HOLSEA sea-level observations must have locations specified to within 2 km; age in calendar years before present; measured or reasonably estimated elevation; and indicative meaning composed of reference water level and indicative range, which respectively define where the indicator formed relative to tidal levels and the 95% confidence range that the indicator occupied[50]. Beyond these criteria, standardized data have an array of additional metadata, including comprehensive estimation of and justification for elevation, age, and inferential uncertainties[22]. Preference in selecting non-HOLSEA papers was given to regions not represented in the HOLSEA database and to data calibrated with IntCal20/Marine20/ShCal20[51–53]; no data were recalibrated for this study. RSL observations from Greenland, Canada, Northern New England, Fennoscandia, British Isles, and Antarctica are excluded from this analysis because of their sensitivity to local mantle viscosity, which limits their utility for GMSL and ice volume inference. RSL observations from locations with known tectonic activity were either not included in the HOLSEA database or were flagged in HOLSEA compilations as being tectonically influenced. We exclude the latter data from this analysis. RSL observations are distributed globally, with the highest data density in Europe, the US, Australia, and Southeast Asia, and data gaps along the West African coastline and in Alaska, Siberia, and the Middle East (Fig. 1). RSL data range in age from 24,295 ka to 1850 CE, and consist of 6664 index points and 3589 limiting points. A limited (n < 50) number of dated samples were described by multiple HOLSEA-standardized papers. For each of these duplicated samples, we chose the interpretation with the most thorough uncertainty assessment, or, when uncertainty assessments were similar, the interpretation from the more recent paper.

### Constructing the ensemble of relative sea level predictions

The sea-level observational dataset assembled for this study is compared to spatiotemporal RSL fields produced by combining estimates of barystatic and thermosteric sea-level change. Predictions of barystatic sea level ($h_b$), defined as the changing proportion of water stored on land and in the ocean[54], are produced by an ensemble of GIA models. Thermosteric sea-level change ($h_\theta$), defined as the temperature-driven expansion or contraction of the global ocean volume divided by the ocean surface area[54], is derived from proxy reconstructions of global mean ocean temperature[55].

The GIA models solve self-consistently for the sea-level equation that accounts for the migration of shorelines, including deformation,

feedbacks into Earth's rotation axis, and gravitational effects in addition to the barystatic effects of ice sheet mass redistribution[4,56]. For the ensemble, we pair various ice thickness histories with a suite of Earth structures. We assume that the elastic structure of Earth's interior follows PREM (Preliminary Reference Earth Model)[57]. For the viscous structure, we vary the elastic thickness of the lithosphere (71 and 96 km), upper mantle viscosity (2, 3, 4, and $5 \times 10^{20}$ Pa S), and lower mantle viscosity (3, 5, 7, 8, 9, 10, 15, 20, 30, 40, and $50 \times 10^{21}$ Pa s). These parameters accord with the range of viable solid Earth structures found by previous RSL data-GIA model comparisons to fit the mid- to low-latitude regions considered here[6,8]. Weaker solid Earth structures, such as are found beneath West Antarctica[58], were not included, as RSL data in far-field locations have not been found to fit GIA-based RSL predictions produced using these structures and are also sensitive to the strong viscosities beneath the Laurentide and Eurasian ice sheets[59,60].

Global ice-sheet reconstructions are constructed by assembling all combinations of 4 Laurentide, 4 Eurasian, 6 Greenland, 1 Patagonian, and 278 Antarctic Ice Sheet histories, then pairing each combination with one of 200 mountain glacier scenarios. Northern Hemisphere ice-sheet reconstructions used include the ANU[61–63], ICE-6G[9], GLAC1D[64,65], and PaleoMIST[7] models. The Huy3[66] and VAR[67] models are included as additional Greenland Ice Sheet reconstructions because of their modest minimum mid-Holocene volume. The Patagonian Ice Sheet history from PaleoMIST is included in all models.

Antarctic ice histories used include 256 Parallel Ice Sheet Model (PISM) ensemble members from Albrecht et al.[15] and an additional 22 histories from Albrecht et al.[24] chosen because they reach a volume smaller than present during the Holocene. A few of the Northern Hemisphere ice sheets used here were either optimized to fit a limited subset of the far-field RSL data we use (ANU) and/or a single solid Earth structure used in our ensemble (ANU, PaleoMIST). However, the vast majority of our global ice histories are composed of ice sheet reconstructions not optimized to either of those constraints. All of the ice histories used here already include a glacial phase (commencing at 80 ka or earlier) except for ICE-6G, the GLAC-1D Eurasian Ice Sheet, the ANU Laurentide Ice Sheet, and the VAR Greenland Ice Sheet. For ICE-6G, a global glaciation phase between MIS-5a (80 ka) and the Last Glacial Maximum (LGM, 26 ka) is constructed to match a GMSL curve based on RSL observations and $\delta^{18}$O records from benthic foraminifera[68]. Glacial ice configurations are assumed to be identical to postglacial geometries with the same GMSL value. Next, ice volumes are calculated for the pre-LGM ICE-6G Eurasian, Laurentide, & Greenland Ice Sheets. These Eurasian, Laurentide, and Greenland Ice Sheet volume histories are then used to construct pre-LGM GLAC-1D, ANU, and VAR ice-sheet histories by matching the glacial histories to post-LGM GLAC-1D, ANU, and VAR ice-sheet configurations with the same volume. All GIA simulations are run from 80 ka to present; for the vast majority of sites, Holocene relative sea level is insensitive to loading history prior to 80 ka, e.g. at 6 ka, RSL at >98% of sites varied by < 0.05 m depending on whether simulations started at 80 ka or 122 ka.

Mountain glacier ice volumes are reconstructed for the past 80 ka. Spatiotemporal estimates of temperature anomalies from 24 ka to present relative to 1850 are taken from the Holocene Data Assimilation DA[38], and Last Glacial Maximum Reanalysis LGMR[37] products. A 200-member paleotemperature ensemble is constructed by pairing 100 random samples from the Holocene DA (0–12 ka) with 100 samples from the LGMR (12–24 ka), then combining those 100 postglacial temperature histories with an additional 100 random samples from the LGMR ensemble (0–24 ka). Temperature ensemble members are linearly interpolated to a degree 256 Gauss-Legendre grid. An existing scaling relation of the equilibrium mountain glacier volume response to global mean temperature changes[69] is expanded to cover −5 to +4.5 °C using Gaussian process regression with a Matérn 3/2 kernel and a linear prior (see Supplementary Fig. 8). This scaling relation is

mapped onto the paleo-temperature ensemble to create a spatio-temporal mountain glacier scaling field. Early industrial mountain glacier mass and area estimates (1901)[70] are converted to volume assuming an ice density of 920 kg $m^{-3}$, then multiplied by the scaling field to produce a time- and space-varying ensemble of mountain glacier volumes 24–0 ka. Glacier volumes are assumed to linearly increase between 80 and 24 ka. Though this assumption elides the details of mountain glacier volumes prior to LGM, the volumes are so small that this choice should not affect the Holocene GMSL inference. A random sample from the mountain glacier ensemble is added to each ice history.

The mountain glacier ensemble produced here accords within uncertainties with quantitative estimates of Holocene mountain glacier contribution to Common Era sea level, which suggest −0.9 ± 2.1 cm of glacier contribution to GMSL from 1800 to 1850[71] and a maximum of 8 ± 1.5 cm of glacier contribution at ~900 CE relative to 1850 CE[72]. It also agrees with more qualitative assessments of minimal mountain glacier volumes in the early-mid Holocene followed by a readvance to the Little Ice Age maximum[40,73]. The procedure outlined above assumes that Holocene mountain glaciers are in equilibrium with local temperature. While the mountain glacier response to changing climate depends on glacial geometry and local climate conditions, glacial volume in most regions lags glacier length by 30 to ~200 years[74] and glacier length in turn lags temperature by 50–200 years[75]. These lags are similar in magnitude to the 200 yr temporal resolution of our model and are based on measurements from a small fraction of all mountain glaciers[74]. Our assumption of mountain glaciers being in equilibrium with temperature is therefore likely a simplification but one that appears appropriate given the temporal resolution of this study. Combining all reconstructions yields 26,784 ice-sheet histories (ice-sheet ensemble), which, when paired with the 88 different Earth structures, results in 2,356,992 RSL fields (GIA ensemble). All ice models are linearly interpolated from a polar stereographic grid onto a Gauss-Legendre grid of degree 256, which represents a spatial resolution of ~1 degree; this interpolation had minimal impact on the volumes of the ensemble.

GIA calculations are performed at this resolution. Ice volume changes used for the GMSL reconstruction are defined as exclusively ice above floatation following ref. 76 Sections "Results", "Contextualizing modern sea level rise", and "METHODS"; Antarctic ice volume changes are defined as inclusive of ice above and below floatation.

Thermosteric sea-level change is derived from a mean ocean temperature reconstruction 25–0 ka[55] using a linearized equation of state[77]:

$$h_\theta = \alpha \Delta T h_o \qquad (1)$$

where $\alpha$, the thermal expansion coefficient, is $1.7 \times 10^{-4}$; $h_o$, the average depth of the ocean, is 3688 m; $\Delta T$ is the change in mean ocean temperature; and $h_\theta$ is ocean thermal expansion. The thermosteric sea-level estimates were modeled using a Gaussian process regression with a Matérn 3/2 kernel (see Supplementary Fig. 7A). Thermosteric sea level during the Holocene reaches a maximum median value of 0.05 (−0.13, 0.26, 90% credible interval) m above present at 5 ka and remains within 10 cm of present values throughout the Holocene. Using a higher-order Taylor expansion[78] yielded results that differed by less than 1 mm over the Holocene and between 1 and 20 mm during the deglaciation. Random samples drawn from the thermosteric posterior were then added as a spatially uniform, time-varying field to the GIA ensemble before these fields were compared to sea level data (Supplementary Fig. 7B). The inclusion of thermosteric effects, as well as of mountain glaciers, was found to have a minimal effect on the posterior. This suggests that the model is not sensitive to factors such

as thermal expansion and glacier volumes that likely dominated centennial-scale GMSL variability over the last few thousand years.

## Statistical analysis algorithm

We estimate Holocene GMSL by conditioning the GIA ensemble on the RSL database to derive a probabilistic posterior. Statistical analysis is performed on datasets composed of (1) all data, (2) only HOLSEA standardized data, (3) only HOLSEA standardized index points, (4) only index points, and (5) a synthetic dataset (see Section "Contextualizing modern sea level rise"). Because each of these data (sub)sets are analyzed in the same way, this section will for simplicity refer to a singular 'RSL database' in order to describe the algorithmic design.

We group observations from the RSL database by geographic location, using a site size of 5 degrees lat/lon (see Fig. 1 for data locations and Supplementary Fig. 9A for sites). Grouping is performed to account for geographic clustering of data; each site receives equal weight in the following misfit analysis. Varying site size by two degrees was found to change the posterior GMSL median <0.02 m over the last 6 kyr, <1 m between 6 and 8 ka, and 1–4 m between 8 and 11.7 ka, and the posterior Antarctic Ice Sheet median by <0.02 m over the last 7 kyr and <0.3 m between 7 and 11.7 ka−differences that are much smaller than the posterior uncertainty.

A fitness score is derived for each sea-level index and limiting point by comparing them to a member of the GIA ensemble via a weighted residual sum of squares (WRSS) calculation following Creel et al.[79] and similar to Briggs and Tarasov[80], which accounts for elevation and age uncertainties in both index and limiting points:

$$\mathcal{W}_{nm} = \begin{cases} \left(\dfrac{2r^t_{nm}}{\epsilon^t_n}\right)^2 + \left(\dfrac{2r^y_{nm}}{\epsilon^y_n}\right)^2 & c_n = 0 \\[2ex] \left(\dfrac{2r^t_{nm}}{\epsilon^t_n}\right)^2 - 2\ln\left(\dfrac{1}{2} + \dfrac{1}{2}\,erf\left(c\,\dfrac{r^y_{nm}}{\epsilon^y_n}\right)\right) & c_n \neq 0 \end{cases} \qquad (2)$$

$\mathcal{W}_{nm}$ is the WRSS for datapoint $n$ and GIA ensemble member $m$, $r^y_{nm}$ and $r^t_{nm}$ are the residuals in sea level and time, respectively, between datapoint $n$ and GIA ensemble member $m$, and $\epsilon^y_n$ and $\epsilon^t_n$ are the standard deviation of the error in observed sea level and observation time, respectively. Uncertainties are assumed to be independent and normally-distributed, and are derived from the original publications or HOLSEA-standardized compilations. Further, $c_n = 0$ when the observation $n$ is a sea level index point, $c_n = -1$ if the datapoint is marine limiting and $c_n = 1$ if the datapoint is terrestrial limiting. In comparing data to GIA ensemble members, data are compared to RSL curves at the nearest grid point, which for all data is within 50 km. A chi-squared value, $\chi^2_{ms}$, is calculated by taking the mean of WRSS scores for each GIA ensemble member $m$ at each site $s$:

$$\chi^2_{ms} = \frac{\sum_{n=1}^{N}(\mathcal{W}_{nm} \cdot \delta_{ns})}{\sum_{n=1}^{N} \delta_{ns}} \qquad (3)$$

where N is the number of observations in the RSL database. $\delta_{ns} = 1$ if datapoint $n$ is in site $s$, otherwise $\delta_{ns} = 0$. For the next step we consider that each GIA ensemble member $m$ can be described by a combination of ice model $i$ and Earth structure $e$, i.e. $\chi^2_{ms}$ can be written as $\chi^2_{ies}$. We next calculate the best possible misfit value for a given ice history and site by choosing the Earth structure that minimizes $\chi^2_{ies}$:

$$\chi^2_{is} = \min_{\forall e}(\chi^2_{ies}) \qquad (4)$$

This procedure assumes that the best fit to the data is obtained for the Earth structure and ice model that is closest to the true one. Note that different 1D Earth structures can be appropriate for different sites given the 3D nature of Earth's viscosity[81]. Because GIA calculations that consider only ice above floatation produce viscosity-dependent GMSL estimates, each $\chi^2_{is}$ has a distinct GMSL curve, $GMSL_{is}$. $GMSL_{is}$ is calculated following ref. 76 Sections "Results" (ice above floatation)

and 4 (bedrock changes). For each ice model, we then take the mean of $\chi_{is}^2$ over all sites $S$, which results in a misfit value for each ice reconstruction:

$$\chi_i^2 = \left( \frac{\sum_{s=1}^S \chi_{is}^2}{S} \right) \qquad (5)$$

This statistic represents the overall fit of a given global ice-sheet history to the RSL database.

We also compute a global mean sea level curve for each global ice-sheet history weighted by $\chi_{is}^2$ metrics:

$$GMSL_i = \left( \frac{\sum_{s=1}^S (GMSL_{is} + h_{\theta s}) \cdot \frac{1}{\chi_{is}^2}}{S} \right) \qquad (6)$$

where $h_{\theta s}$ is a random draw from the thermosteric contribution posterior (see Supplementary Fig. 7). Note that $\chi_{is}^2$ metrics are inverted to convert them to weights.

An additional statistical analysis is employed to estimate the cross-correlation between mean annual and summer air temperatures from the West Antarctica Ice Sheet Divide ice core[2,29] and median Antarctic ice volumes (this study). We apply detrending windows of between 400 and 3000 years to the median Antarctic ice volumes and estimate cross-correlation following refs. [82,83], with significance estimated via a 500 member ensemble analysis.

## Data−model misfit and viscosity inference

Data-model misfit metrics for each sea-level data site and ice model reveal how well sites fit the ice-sheet ensemble. In contrast to most locations, misfits in the Yellow Sea, Vietnam, Timor-Leste, and Namibia are disproportionally large (Supplementary Fig. 9A). A disproportionate misfit indicates that no combination of ice history and solid earth structure produced RSL curves that fit the observations at that site and accord with the full database in terms of ice history. These misfits suggest the influence of local processes such as tectonics (e.g. Timor-Leste), deltaic subsidence (e.g. Yellow Sea), or local sediment dynamics (e.g. Namibia, Cameroon). We used data from two sources: compilations following agreed-upon community standards[22,84], and published indicators not yet compiled to these standards. Standardized RSL observations are found to fit the ice-sheet ensemble 41% better than un-standardized observations.

We find that best-fitting Earth structures are broadly coherent at both nearfield and farfield sites. Our algorithm is insensitive to upper mantle viscosity: the vast majority of sites are best fit by upper mantle viscosities around $3.5 \times 10^{20}$ Pa s, with modestly higher viscosities near the peripheral bulges of the Laurentide and Eurasian Ice Sheets and lower viscosity in Patagonia, the US West Coast, and the Gulf of Mexico (Supplementary Fig. 9B). This aligns well with the preferred global viscosity structure of ref. [6] and the preferred upper mantle viscosity for Southeast Asia of ref. [8], but stands in contrast to the strong upper mantle inferred by ref. [59] for the Caribbean. Lower mantle viscosities are weakest in the intermediate field (Mediterranean, US West Coast, Caribbean) and variable in the far-field. Weak lower mantle viscosities in the Caribbean accord with ref. [59], while the bi-modal distribution of weak ($3–10 \times 10^{21}$ Pa s) and strong ($-5 \times 10^{22}$ Pa s) lower mantle viscosities that we infer for Southeast Asia and Australia, respectively, accords with ref. [6], which relies heavily on data from that region. Lithospheric thickness varies regionally from high values (>90 km) around Maine, Central Europe, and Indonesia to low values (<75 km) for Argentina, the Eastern Mediterranean, South Africa, the UK, Western Russia, US West Coast, and southern India (Supplementary Fig. 9D). These patterns accord remarkably well with maps of lateral variation in lithospheric thickness[85], which also place thick lithosphere

in China and in Indonesia and weak lithosphere in Western Russia, and around the UK.

That our viscosity inferences broadly accord with viscosities inferred by prior GIA-based GMSL studies increases confidence in the viability of our method for inferring GMSL. However, we caution against over-interpretation of these viscosity maps and others based strictly on GIA models that do not include lateral variations in mantle viscosity and rely on Maxwell Earth structures. RSL is sensitive to Earth structure both locally and beneath areas of ice mass change and the degree of depth-sensitivity varies between near- and far-field sites[86]. Additionally, apparent viscosity structure as sensed by RSL data depends on the timescale of deformation, which implies that sites with predominantly older (e.g. early-mid Holocene) RSL data may sense a different viscosity structure than sites where younger (e.g. Common Era) data dominate[81,87,88]. Future efforts to invert RSL observations for viscosity structure should apply more nuanced tools such as adjoint sensitivity kernels[86].

Separate from our analysis, goodness-of-fit information for 256 of the PISM Antarctic Ice Sheet simulations used in this study was calculated by Albrecht et al.[15,23]. These fitness metrics assess how well the PISM runs align with six types of present-day observational constraints and three types of paleo-constraints. Present-day constraints include grounded area, ice shelf area, ice thickness, grounding-line location, uplift rates, and grounded surface ice speed; paleo-constraints include grounding line position at LGM as well as cosmogenic-derived surface elevation and ice extent between LGM and present[15]. We perform identical fitness assessments for the additional 23 PISM Antarctic simulations published in[24], all of which include an Antarctic Ice Sheet smaller than present during the Holocene. These fitness metrics $P_i$ are assigned to each ice ensemble member based on which PISM Antarctic reconstruction it includes.

For our prior GMSL distribution, we prescribe the probability that GMSL at 6 ka was between −10 and +2 m to be uniform and the probability that it was lower than −10 m to decrease asymptotically to zero by ~−15 m. These values are a conservative bracket around the range of values (−3.5 to +0.5 m) chosen by the IPCC AR6 assessment report[21]. To create this uniform prior at 6 ka, we calculate a weighting factor $U_i$ for each ice-sheet model (Supplementary Fig. 3A−C).

Putting it all together, each ice-sheet ensemble member $i$ has a global mean sea level curve, $GMSL_i$; an associated weighting factor $U_i$, which produces a uniform prior at 6 ka; and a statistical analysis factor (sum of the inverse of fitness scores derived from RSL observations, $\frac{1}{\chi_i^2}$, and PISM model weights, $P_i$), which captures how well this ice-sheet history fits non-RSL observations. PISM model weights are generated from comparison of model outputs to six types of non-RSL modern observations and three types of paleo evidence[15,23]. We use the inverse of $\chi_i^2$ fitness scores in order that models with higher scores are weighted more heavily. Note that for the statistical analysis factor we choose to sum the two scores rather than multiplying them because summation allows an overall good score to be obtained from a good fit either to sea-level observations or to ice constraints, but does not require both. This approach, which is more conservative than requiring that modeled RSL fit both observational datasets well, produces our final ice model weights $w_i$:

$$w_i = \frac{\left( \frac{1}{\chi_i^2} + P_i \right) * U_i}{\sum_{i=1}^I \left( \frac{1}{\chi_i^2} + P_i \right) * U_i} \qquad (7)$$

Note that the denominator serves to normalize the final weights such that they sum to 1 and that $U_i$, $\chi_i^2$, and $P_i$ are separately normalized to sum to one prior to combination. The weights are multiplied with the $GMSL_i$ curve of each ice-sheet ensemble member to produce a posterior GMSL distribution. Results of this study are reported as having a 'credible' interval because models have an associated

likelihood; uncertainty estimates from studies not produced via Bayesian methods or without associated likelihoods are reported as having a 'confidence' interval.

The fitness score for each site, visualized in Supplementary Fig. 9A, is calculated as the average of fitness scores (equation (4)) weighted by the associated ice model's global fitness score (equation (5)). Optimal Earth structures are computed for each site as the mean of the Earth structures identified in equation (4) weighted by the linear combination of local data-model misfit $\chi^2_{is}$ and global ice model weight $w_i$:

$$\chi^2_s = \frac{\frac{1}{\chi^2_{is}} + w_i}{\sum_{i=1}^I \frac{1}{\chi^2_{is}} + w_i} \qquad (8)$$

Note that the denominator serves to normalize the final weights such that they sum to 1 and that $\frac{1}{\chi^2_{is}}$ and $w_i$ are separately normalized to sum to one prior to combination. The combination of $\frac{1}{\chi^2_{is}}$ and $w_i$ preferences local information while also including spatial covariation between sites.

The relative performance of HOLSEA and non-HOLSEA databases is compared by assigning each observation the site-specific fitness value from Supplementary Fig. 9A, then comparing the average fitness scores of standardized and un-standardized observations. We also explore the sensitivity of our algorithm to subsets of the RSL database, including using only index points, using only index points standardized according to agreed-upon community standards[22], and using only standardized data including both index points and limiting points (Supplementary Fig. 11).

Fitness scores are used to estimate the characteristics of Holocene GMSL and Antarctic volume in two distinct ways. First, the fitness scores are used to create posterior distributions of time-varying GMSL and ice volume change characterized by their medians and credible intervals (Fig. 2B, C). Second, the peak value and its timing for each $GMSL_i$ curve (and minimum value of each Antarctic volume curve) is identified. Many $GMSL_i$ curves peak at present; others reach a peak earlier in the Holocene. We weigh these peaks (or minima) using the $\chi^2_s$ fitness scores. To obtain a continuous distribution, we then assign each peak 0.2 m of elevational uncertainty and 0.5 kyr of age uncertainty, both normally distributed, and combine these probability distributions of each ice history into a joint probability density distribution (Fig. 3). This procedure is performed for both prior and posterior GMSL (and Antarctic volume, 3A, B, D, E); the ratio of prior to posterior distribution of GMSL peaks (or Antarctic ice minima) represents the amount of information that RSL data and nearfield Antarctic constraints contribute (Fig. 3C, F).

## Tests to demonstrate model performance

Synthetic and empirical tests are performed to assess the skill of the statistical analysis algorithm in estimating GMSL and reproduce RSL. To test the algorithm's skill at estimating GMSL, we select a subset of ice histories ($n = 9$) that represent the full range of Holocene GMSL scenarios and remove them from the ice-sheet ensemble. Spatio-temporal RSL fields are calculated from each ice history using a lithospheric thickness of 71 km, an upper mantle viscosity of $2 \times 10^{20}$ Pa s, and a lower mantle viscosity of $40 \times 10^{20}$ Pa s. All other GIA ensemble members with this viscosity structure are also removed from the ensemble. In addition to these 1D GIA realizations, we also include one RSL field produced by GIA calculations using laterally-varying viscosity structure and the ICE-6G global ice history, the details of which are described by Austermann and colleagues[89].

Each of the 9 1D and 1 3D RSL fields are sampled at the locations and ages of the 10,253 RSL observations and assigned uncertainties identical to those of the data. This procedure produces 10 synthetic

RSL datasets. We infer a posterior GMSL using each of these synthetic datasets and the approach described in the previous section, modified such that only weights derived from RSL sources are used. We then compare the resulting GMSL to the GMSL curve associated with the ice history that produced the synthetic data. For each time step, 'coverage' is calculated as the percentage of estimated GMSL curves whose credible interval intersects the 'true' GMSL curve; the coverage test is passed if this percentage approximates the credible interval, e.g. if around 95% of comparisons pass for a 95% credible interval. Coverage for 1D and 3D simulations is shown in Supplementary Fig. 5D-F for differing credible intervals. Synthetic tests with a 95% and 90% credible interval have 100% coverage between 11.5 and 6 ka and 75–90% coverage in the late Holocene; the failing models are generally those whose GMSL is higher than 1 m above present through the mid-late Holocene. Assuming a 66% credible interval yields 60% coverage or greater for all the Holocene, with failure concentrated around late Holocene high and low GMSL scenarios. That the model is able to reproduce all but the most extreme GMSL scenarios for both 1D and 3D simulations increases confidence in the application of our algorithm to estimate Holocene GMSL.

We next demonstrate our algorithm's efficacy at producing RSL curves that match observations within uncertainties. A selection ($n = 20$) of sites $s$ of size 5 degrees lat/lon are chosen in regions of high data density, and for each site and ice model the RSL curve $RSL_i$ which best fit nearby data, i.e. the curve with the minimum $\chi^2_{is}$, is selected. These curves are merged using the $\chi^2_i$ weights assigned to the curves' associated ice models in order to produce a posterior RSL curve $RSL_s$:

$$RSL_s = \sum_{i=1}^I RSL_i * \frac{\frac{1}{\chi^2_i}}{\sum_1^I \frac{1}{\chi^2_i}} \qquad (9)$$

This procedure balances local information−included through the selection of $RSL_i$ curves via $\chi^2_{is}$ minimization−with the global information contained in the $\chi^2_i$ weights.

Data-model comparisons are plotted in Supplementary Figs. 13–16 for sites where more than four index points fall within one degree latitude/longitude of the site location−a cutoff chosen to minimize plotting artifacts. Also for this reason, only RSL data within one degree latitude/longitude of the center of each site are plotted.

Comparisons between RSL data and modeled RSL support the aggregated misfit calculations shown earlier (see "Methods" Section "Contextualizing modern sea level rise" and Supplementary Fig. 9). For instance, higher misfit scores in Southeast Asia correspond to systematic misfit trends in Supplementary Figs. 14F (Vietnam) and 15E (Hainan and Shanghai, China) and align with the lack of correction for local subsidence in these data (e.g. ref. 90). Similar misfits due to lack of subsidence correction of deltaic index points occur in Supplementary Fig. 14E (c.f. ref. 91). All misfitting index points from these regions that plot outside the credible interval of our RSL reconstruction are from papers not standardized following the HOLSEA protocols. This misfit pattern motivates future work to standardize RSL data in these regions. Beyond Supplementary Figs. 15E / 14F, modeled RSL curves fit virtually all RSL data within uncertainties save in Supplementary Fig. 13E, F (New Jersey/North Carolina). There RSL data consistently plot above modeled RSL in the early-mid Holocene, a trend that mirrors the offset between GIA models and RSL data prior to 4 ka noted by ref. 92, which they attribute to inaccuracies in ice histories−an assessment we support.

## Comparison to future sea level

The IPCC Sixth Assessment Report projects that processes that can be modeled with medium confidence will contribute 0.44 m (0.32–0.61 m, at least 66% probable range) to GMSL in a low emissions scenario (SSP1-2.6) and 0.68 m (0.55–0.90 m) in a higher

emissions scenario (SSP3-7.0). Concerning the Antarctic Ice Sheet, the IPCC projects that processes that can be modeled with medium confidence will contribute 0.11 m (0.03–0.27 m, at least 66% probable range) to GMSL under SSP1-2.6 and 0.11 m (0.03–0.31 m) under SSP3-7.0. Projections for a low-likelihood, high-impact future scenario (SSP5-8.5) that incorporate processes about which there is low confidence place 83rd percentile GMSL projections at 1.61 m and the 83rd percentile Antarctic contribution at 0.56 m.[21] These values are relative to a 1995–2014 baseline period, while our GIA calculations are relative to early industrial (1850) values. We extend the IPCC baseline to 1850 with historical GMSL and Antarctic ice volume estimates from the IPCC AR6 (1950 to 2020, 1980 to 2020) and (ref. [93], 1850 to 1950; ref. [94]). Because no estimates for Antarctic ice volume exist between 1850 and 1900, we extrapolate to 1850 the linear Antarctic contribution of 0.05 ±0.04 m that ref. [94] adopted between 1900 and 1980.

This produces an estimate of 0.61 m (0.50–0.79 m) of GMSL rise and 0.13 m (0.05–0.29 m) of Antarctic Ice Sheet contribution between 1850 and 2100 for SSP1-2.6, 0.85 m (0.72–1.07 m) and 0.13 m (0.05–0.33 m) for SSP3-7.0, and 1.05 m (median) to 1.78 m (83rd percentile) (GMSL) and 0.21 (median) to 0.58 (83rd percentile) for SSP5-8.5 (Fig. 4). We calculate the probability that these future sea level and ice volume projections exceed our Holocene GMSL and Antarctic ice volume reconstructions by computing the fraction of the 20,000 posterior samples from each of the seven IPCC AR6 GMSL workflows[95] that exceed samples drawn from our Holocene reconstructions. Each IPCC workflow consists of a set of sea-level components—e.g. the sea level contribution of thermosteric effects or the Antarctic Ice Sheet—that were combined in order to create a probabilistic estimate of GMSL.[95] A probability envelope is produced following the IPCC-AR6 'p-box' framework[96]. For each emission scenario, the highest and lowest exceedance probabilities at each time step are chosen; this envelope represents the uncertainty in the exceedance probability estimate (Supplementary Fig. 12). Only the lowest (highest) exceedance probabilities are shown for the low-probability SSP1-1.9 (SSP5-8.5, low confidence) pathways, as these pathways represent outer boundaries on the likely amount of future sea-level rise.

## Data availability
The data generated and used in this study and code necessary to produce the model results have been deposited in the Zenodo database under accession code https://doi.org/10.5281/zenodo.7986159.

## Code availability
Code to produce GIA models is available at https://github.com/jaustermann/SLcode/. Scripts for processing GIA outputs and producing plots are available at the Zenodo link above.

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

## Acknowledgements

This work was supported by National Science Foundation grants EAR-2002352 (RC, JA), OCE-2002437 (REK) and EAR-2148265 (REK). T.A. was supported by the Deutsche Forschungsgemeinschaft (DFG) in the framework of the priority program "Antarctic Research with comparative investigations in Arctic ice areas" by grants WI4556/2-1 and of the PalMod project (FKZ: 01LP1925D), supported by the German Federal Ministry of Education and Research (BMBF) as a Research for Sustainability initiative (FONA). We acknowledge computing resources from Columbia University's Shared Research Computing Facility project, which is supported by NIH Research Facility Improvement Grant 1G20RR030893-01, and associated funds from the New York State Empire State Development, Division of Science Technology and Innovation (NYSTAR) Contract C090171, both awarded April 15, 2010. The authors acknowledge PALSEA, a working group of the International Union for Quaternary Sciences (INQUA) and Past Global Changes (PAGES), which in turn received support from the Swiss Academy of Sciences and the Chinese Academy of Sciences. Thanks to Matteo Vacchi, Sarah Shackleton, Christopher Piecuch, Darrell Kaufman, and Erica Ashe for helpful correspondence, and to Jonathan F. Donges for support on the cross correlation analysis.

## Author contributions

R.C., J.A., and R.K. conceived and designed the research; R.C. wrote and executed the GMSL algorithm with guidance from J.A. and R.K.; J.A. wrote the GIA code; R.C. and N.K. compiled the data; T.A. supplied Antarctic Ice Sheet simulations and computed Antarctic fitness scores; R.C. drafted figures and wrote the original draft with help from J.A. and R.K.; R.C., J.A., R.K., T.A., J.K., and N.K. contributed to manuscript review and editing.

## Competing interests

The authors declare no competing interests.

## Additional information

**Peer review information** : *Nature Communications* thanks Marc Hijma and the other, anonymous, reviewer(s) for their contribution to the peer review of this work. A peer review file is available.

