## [Transparent Peer Review file · Nature Communications]

Global mean sea level likely higher than present during the Holocene

Corresponding Author: Dr Roger Creel

Version 0:

Reviewer comments:

Reviewer #1

(Remarks to the Author)

[Note that this review will only address how well the authors responded to my original comments on the earlier version of this paper (for which I was Referee 4.) I have experience with statistical analysis but not with paleoclimate or global mean sea level.]

I appreciate the authors' effort to address each of my comments. I am very nearly satisfied with the changes made; I just have two small comments.

First, in the new Eqn (8), it seems again that there is a typo and the chi-squared symbol should be replaced with the inverse (as was modified in Eqn 7.)

Second, on lines 426 and 467, did the authors mean to refer to the Extended figures?

Reviewer #2

(Remarks to the Author)

I have read the rebuttal to my comments and suggestions, and compliment the authors on their sound, detailed and thorough reply. They have addressed all my points satisfactorily. I especially appreciate the newly constructed Extended Data Figure 11 since it clearly demonstrates the importance of using the HolSea-standard and limiting datapoints.

Reviewer #3

(Remarks to the Author)

This paper describes the Global Mean Sea Level (GMSL) above the present (pre-industrial) level, in contrast to other previously reported values. In this manuscript, Creel performed a statistical analysis of sea level using "20 times more" sea level observations available in the literature. They concluded that 0.3 to 1 m above today's GMSL was reached 3,000 years ago. Antarctic ice retreat was also reported to lag temperature changes by up to 250 years. I appreciate the authors' extensive exercises using a geophysical model to understand various physical processes, including glacio-isostatic adjustments (GIA), thermosteric expansions, and ice sheet dynamics. Although the novel aspects of this study are the increased number of data, as the authors stated the uniqueness of this paper, I am not convinced that the approaches can solve the fundamental issue behind the topic, namely the newly proposed GMSL can uniquely eliminate the "apparent overshoot" of GMSL as small as less than 1 m. Therefore, I do not think the manuscript has crossed the hurdle to be published in this journal.

Previous studies based on various sea level observations combined with GIA modelling have never exceeded the Holocene GMSL compared to the present level. This trend is also consistent with the history of global mean temperature reconstructed by IPCC climate models, i.e. modern temperature is the highest in the last 11,700 years. The widely accepted view for the GMSL is that it reached about -3 m between 7,000 and 6,000 years ago, coinciding with the disappearance of the Laurentide

Ice Sheet (LIS). During the next 3,000 years, the GMSL rose by 3 m until 3,000 years ago. This Holocene melting is widely attributed to the melting of the Antarctic and Greenland ice sheets. These GMSL reconstructions can explain, within conservative observational uncertainties, most of the sea level observations found in different areas of the world. Individual sea level observations cannot be treated in the same way because the biological tolerance to environmental conditions varies depending on the type of indicator. For example, the modelled sea level should not be both below corals and above terrestrial limiting indicators (i.e. freshwater indicators). Their plots in Figure A1, shaded envelopes, tend to be plotted higher than the marine limit indicators. This can also be seen in the extended Figure 1, where the 'new' GMSL is up to 10m higher than previously reported curves, including the early Holocene. This suggests a potential bias in the current method. Tidal ranges often influence sea level indicators, and tectonic effects cannot be ignored even during the Holocene, contrary to what the authors state. It is therefore really difficult to distinguish these to resolve 0.5m GMSL variations.

In terms of the timing of melting, the current paper concludes that the GMSL peak does not coincide with temperature changes. They also suggest that Antarctic melting has lagged behind changes in Antarctic air temperature. If this were true, it would be important to understand the sensitivity of the Antarctic ice sheet to climate. However, it is widely recognised that the age model for Antarctic melt observations has more than 10-20% uncertainties associated with the complications of radiocarbon dating, including the age of marine reservoirs (e.g. 1000 year age differences or so) or exposure ages (e.g. 10-20% of age differences due to production rate estimates and other factors). Thus, their conclusion regarding the timing of the Antarctic melt is not convincing because it was derived without considering these factors.

Instead of taking a "big data approach", I strongly recommend the authors to show region by region comparisons between the currently proposed GMSL-based predicted sea level and observations. This will be a very important contribution to the community, but it will require much more space offered by Nature Communications.

Minor points:

1) Some minor points are that the figures do not clearly show the magnitude of the higher mid-Holocene GMSL due to the tick marks and the size of the figures. The plots should be enlarged to show the differences between the GMSL models, especially in the mid to late Holocene. The plot also ignores significant figures as the authors argue that their GMSL can resolve less than 0.5m sea level.

2) The thermosteric sea level correction cannot be made precisely, as we also have large uncertainties in the Holocene sea surface temperature using climate models (cf. CMIP). Therefore, the associated uncertainties are also likely to be underestimated.

Version 1:

Reviewer comments:

Reviewer #2

(Remarks to the Author)

I was already satisfied with the previous version and hence still am. The replies to reviewer #3 are again detailed and thorough, well done. I therefore recommend publication.

(Remarks on code availability)

Reviewer #4

(Remarks to the Author)

I have been asked to evaluate the authors' responses to Rev. 3, especially the reviewer's concern that the study's newly proposed GMSL can't uniquely eliminate the "apparent overshoot" of GMSL. (It took me a while to understand what this was referring to since the authors don't use this terminology, but it's basically saying that late Holocene sea level did not exceed 1 m).

The authors do not address this specific question. Instead, they point to the various novelties of the paper, thus addressing the reviewer's earlier comment implying that the only novelty of the paper is the inclusion of new data, and they are correct in doing so. However, regarding the "apparent overshoot" of GMSL, the authors only respond with "We agree with Reviewer 3 that our results don't uniquely constrain GMSL to have been higher than 1m during the Holocene" (although they must mean lower than 1 m, not higher). This is not clearly stated in the paper, yet a key take-home point of the paper (Section 4) is to use this constraint to conclude that "Future GMSL will more likely than not ($P > 0.5$) exceed maximum Holocene GMSL by 2060 under all emissions scenarios (Fig. 4A). By 2150, future GMSL will likely ($P > 0.66$) be higher than peak Holocene GMSL under low emissions (SSP1-2.6) and very likely ($P > 0.9$) higher under high emissions (SSP5-8.5)." I agree with the authors that the strength of the paper is a new and robust estimate of Holocene sea level that in places differs from previous reconstructions, which is to be expected given the improvement in the models and the large increase in data, and in this regard I support publication. However, I think the authors still need to explicitly address Rev. 3's comment about the "overshoot," especially in the context of Section 4.

(Remarks on code availability)

We thank the editor for handling this process and the three reviewers for their careful review of our manuscript “Global Mean Sea Level Higher than Present in the Holocene”. We have revised the manuscript to incorporate the reviewers’ comments and suggestions. Please find below our rebuttal where we address all the reviewer’s comments point-by-point, interspersing the review (in black) with our response (in blue). Note that line numbers in blue refer to the revised version of the manuscript. We also include a version of the manuscript that indicates all changes from the original manuscript.

REVIEWER COMMENTS

Reviewer #1 (Remarks to the Author):

[Note that this review will only address how well the authors responded to my original comments on the earlier version of this paper (for which I was Referee 4.) I have experience with statistical analysis but not with paleoclimate or global mean sea level.]

I appreciate the authors’ effort to address each of my comments. I am very nearly satisfied with the changes made; I just have two small comments.

First, in the new Eqn (8), it seems again that there is a typo and the chi-squared symbol should be replaced with the inverse (as was modified in Eqn 7.)

We agree with Reviewer 1 that this equation has a typo and should have the same inverse as Eqn 7. We have made this change.

Second, on lines 426 and 467, did the authors mean to refer to the Extended figures?

Yes, we did mean to refer to the Extended figures. We have made this change.

Reviewer #2 (Remarks to the Author):

I have read the rebuttal to my comments and suggestions, and compliment the authors on their sound, detailed and thorough reply. They have addressed all my points satisfactorily. I especially appreciate the newly constructed Extended Data Figure 11 since it clearly demonstrates the importance of using the HolSea-standard and limiting datapoints.

We thank Reviewer 2 for their approval!

Reviewer #3 (Remarks to the Author):

This paper describes the Global Mean Sea Level (GMSL) above the present (pre-industrial) level, in contrast to other previously reported values. In this manuscript, Creel performed a statistical analysis of sea level using "20 times more" sea level observations available in the literature. They concluded that 0.3 to 1 m above today's GMSL was reached 3,000 years ago. Antarctic ice retreat was also reported to lag temperature changes by up to 250 years. I appreciate the authors' extensive exercises using a geophysical model to understand various physical processes, including glacio-isostatic adjustments (GIA), thermosteric expansions, and ice sheet dynamics. Although the novel aspects of this study are the increased number of data, as the authors stated the uniqueness of this paper, I am not convinced that the approaches can solve the fundamental issue behind the topic, namely the newly proposed GMSL can uniquely eliminate the "apparent overshoot" of GMSL as small as less than 1 m. Therefore, I do not think the manuscript has crossed the hurdle to be published in this journal.

We thank Reviewer 3 for their assessment of our work. We note that our manuscript enumerates several other ways in which our work is novel beyond the 10- to 20-fold increase in sea-level observations used over previous global mean sea level (GMSL) estimates. We summarize these contributions here:

- We produce the first quantitative estimate of Holocene mountain glacier volume.
- We produce the first quantitative estimate of Holocene thermosteric sea level change.
- Our study is the first GMSL study to use a large ensemble of distinct ice histories (n=26,784) and GIA models (n=2,356,992).
- Our study is the first GMSL study to include marine- and terrestrial-limiting data as data constraints.
- We develop a novel method to approximate the influence of laterally-varying solid Earth structure.
- Our study is the first to quantitatively compare Holocene GMSL to future GMSL change as projected by the International Panel on Climate Change.

We agree with Reviewer 3 that our results don't uniquely constrain GMSL to have been higher than 1m during the Holocene. However, we don't think this should be the threshold for publication in this journal. Instead, we believe that the threshold for publication in this journal should be a best estimate of GMSL during the Holocene based on extensive data and models compared using ground-truthed statistical tests – agnostic to what specific values are reached.

Previous studies based on various sea level observations combined with GIA modelling have never exceeded the Holocene GMSL compared to the present level.

We agree that previous studies that combined GIA modeling with sea-level observations did not conclude that GMSL substantially exceeded the present. We already describe this in the manuscript and provide eight possible reasons for why our results differ. In light of this comment, we've again revisited past work and added even more context as we describe below.

The algorithm employed by the most comprehensive previous study, Lambeck et al. 2014, produced two GMSL solutions: a ‘low viscosity’ solution, and a ‘high viscosity’ solution (referring to choices in lower mantle viscosity). Lambeck and colleagues note that while there is evidence for both Greenland and Antarctic melt in the mid-late Holocene, the available field evidence at that time was not sufficient to quantitatively determine the behavior of mountain glaciers, the Antarctic ice sheet, or Greenland ice sheets after 7 ka. The higher estimate of Lambeck et al. (2014), which is based on the ‘low viscosity’ case and reproduced below, exceeds present levels by 10 to 20 cm starting around 4 ka. Lambeck and colleagues further note that their algorithm is only able to distinguish GMSL variations with amplitudes of more than 10-20 cm within the last 4 kyr. We note that Lambeck et al. overall favor the ‘high viscosity’ case based on a lower GMSL estimate during the last glacial maximum and independent evidence for a significant viscosity increase across the transition zone.

Fig. 1. Global mean sea level estimates from past work and this study. Light grey envelope indicates 90% credible interval.

The most recent GMSL study, Bradley et al. (2016), produces a GMSL curve via comparison of GIA models to far-field RSL data. However, they intentionally do not explore any scenarios where GMSL exceeds the present, thus predetermining that their best-fitting GMSL solution will never exceed present levels.

We argue that though prior studies do not conclude that Holocene GMSL exceeded the early industrial, the Lambeck study provides support for our conclusion that GMSL that exceeded the present in the mid-late Holocene, and the Bradley is agnostic on the issue. We have added text to clarify this point (Line 54-55 and 61-62):

Lambeck et al. (2014) acknowledge that lack of observational evidence precludes independently constraining cryospheric fluctuations between 7~ka and the early industrial.

Lastly, they may have overlooked the possibility of GMSL higher than present because of algorithmic design, for instance by exclusively using ice sheet histories whose Holocene

volume never was smaller than at early industrial (Bradley et al. 2016) or by enforcing a single optimal history even when a second best-fitting curve exists that includes GMSL higher than present (Lambeck et al. 2014, see extended Fig. 1).

To acknowledge the fact that we find it 75% likely that GMSL exceeded pre-industrial during the Holocene, rather than >95% likely, we have changed the manuscript title to “Global mean sea level likely higher than present during the Holocene.”

This trend is also consistent with the history of global mean temperature reconstructed by IPCC climate models, i.e. modern temperature is the highest in the last 11,700 years.

We first note that our study reconstructs Holocene GMSL up to the pre-industrial (1850), not the modern, i.e. 21st century, which we state on line 71 and throughout the text. In that context, we respectfully rebut the assertion that the IPCC reconstructs pre-industrial global mean temperature as the highest in the Holocene. We have reproduced below Figure 2.11a from the IPCC-AR6 along with the associated caption.

Fig. 2. Earth’s surface temperature history with key findings annotated within each panel. (a) GMST over the Holocene divided into three time scales: (i) 12 kyr–1 kyr in 100-year time steps; (ii) 1000–1900 CE, 10-year smooth; and (iii) 1900–2020 CE (from panel (c)). Median of the multi-method reconstruction (bold lines), with 5th and 95th percentiles of the ensemble members (thin lines). Vertical bars are the assessed *medium confidence* ranges of GMST for the Last Interglacial and mid-Holocene Section 2.3.1.1). The last decade value and *very likely* range arises from Section 2.3.1.1.3.

The IPCC adopts the global mean temperature reconstruction of Kaufman et al. (2020), which finds that the median Holocene global mean surface temperature peaked ~0.5°C above present 5,500 years ago. This global mean temperature history would be consistent with GMSL peaking above present. In our manuscript, we note that Holocene GMSL is best compared to Holocene polar temperatures, since the cryosphere responds to local, not global mean temperature.

Kaufman et al. (2021) find that Arctic temperatures peaked higher than present in the mid-Holocene; Antarctic temperatures had a similar maximum in the late Holocene, as we show in Figure 2 (Cuffey et al. 2016). These polar temperature histories are also consistent with Holocene GMSL higher than present.

The widely accepted view for the GMSL is that it reached about -3 m between 7,000 and 6,000 years ago, coinciding with the disappearance of the Laurentide Ice Sheet (LIS). During the next 3,000 years, the GMSL rose by 3 m until 3,000 years ago. This Holocene melting is widely attributed to the melting of the Antarctic and Greenland ice sheets. These GMSL reconstructions can explain, within conservative observational uncertainties, most of the sea level observations found in different areas of the world.

As shown in Fig. 1 of this document and noted in lines 32-38 in the main text, the field of Holocene GMSL research prior to this paper had not reached consensus. Bradley and colleagues (2016) found that GMSL reached -3 m at 6 ka, as did Lambeck and colleagues (2014) in their high viscosity scenario. However, Lambeck and colleagues also found in their low viscosity scenario that GMSL reached -1.5 m by 6 ka, and Peltier and colleagues (2015) model GMSL at 6 ka to be -0.5 m. We agree that these estimates explained most of the far-field sea-level observations used in those studies, which numbered <100 (Peltier et al. 2015), <500 (Bradley et al. 2016), and ~1,100 (Lambeck et al. 2014). Using a 10- to 20-times larger dataset (and additional advances as pointed out above), however, we find that the median of our best-fitting GMSL history exceeds the present by 0.32 cm in the late Holocene, and it is 75% likely that GMSL exceeded the present.

Individual sea level observations cannot be treated in the same way because the biological tolerance to environmental conditions varies depending on the type of indicator. For example, the modelled sea level should not be both below corals and above terrestrial limiting indicators (i.e. freshwater indicators). Their plots in Figure A1, shaded envelopes, tend to be plotted higher than the marine limit indicators.

We thank Reviewer 3 for highlighting that sea-level observations must be treated according to their type and variety. We agree. This information is codified in the indicative meanings, which relate the indicator to mean tide level at the time it formed. The majority of the data we use were standardized following the HOLSEA working group guidelines for rigorous uncertainty quantification. All other data were used as originally reported in published literature, and were screened to exclude data lacking sufficient metadata to verify their quality. Each of our RSL data has a uniquely-defined indicative meaning in addition to a calendar age, location, and other metadata; our data are not all ‘treated in the same way.’

We next respond to Reviewer 3’s points about Figure A1 from the previous rebuttal, which we copy here for reference:

Figure 3. Comparison of median posterior relative sea level (RSL) prediction to RSL data from Fiji (A) and Tuamotu (B). RSL data are from Tan et al. (2023).

First, the data that we were plotting in this figure were not used in our analysis, so their positioning above the median of our modeling results is not unexpected. Second, even had they been included it would still be possible that a posterior RSL predicts a value somewhat inconsistent with a SL data point (whether that's limiting or index point). This is because the GIA model imposes some requirement of smoothness. If data from neighboring locations are "inconsistent" with one another, the model will favor models whose values fall between them. Lastly, we are now addressing this point more broadly by including 20 RSL curves in our supplementary material (see below) that are plotted against data used in the model.

This can also be seen in the extended Figure 1, where the 'new' GMSL is up to 10m higher than previously reported curves, including the early Holocene. This suggests a potential bias in the current method.

Our GMSL curve in the early Holocene is in places up to 10 m higher than Lambeck and colleagues (2014) but more than 5 m lower than Peltier and colleagues (2015), and our uncertainties include all previous estimates. We therefore argue that the differences between our posterior GMSL curve and previous curves are not a reason to suspect bias; this argument is bolstered by the synthetic tests we conducted – and which are shown in Extended Figure 5 and described in the manuscript's Methods section – to demonstrate our model's ability to reproduce a range of known GMSL scenarios.

Tidal ranges often influence sea level indicators, and tectonic effects cannot be ignored even during the Holocene, contrary to what the authors state. It is therefore really difficult to distinguish these to resolve 0.5m GMSL variations.

We agree that tidal ranges often influence sea-level indicators. Tidal range is included in the uncertainty quantification performed as part of the HOLSEA standardization procedure that was applied to the majority of our data; similar assessments are standard for sea-level data and were generally applied by the original authors of the published data we used that were not in HOLSEA format. While tidal ranges are likely not constant through time, the difference between paleo-

tidal range and modern tidal range is likely to be small in the mid- to late- Holocene. This effect is therefore accounted for in our work (rather than ignored as the reviewer states).

We also respectfully disagree that our manuscript states that tectonic effects can be ignored during the Holocene. Though the manuscript did discuss (lines 379-380) locations that may be influenced by local processes such as tectonics, we agree that our manuscript should be more explicit in describing how we deal with data from areas of tectonic activity. We therefore add to the methods this text (Line 242-243):

RSL observations from locations with known tectonic activity were either not included in the HOLSEA database or were flagged in HOLSEA compilations as being tectonically influenced. We exclude the latter data from this analysis.

In terms of the timing of melting, the current paper concludes that the GMSL peak does not coincide with temperature changes. They also suggest that Antarctic melting has lagged behind changes in Antarctic air temperature. If this were true, it would be important to understand the sensitivity of the Antarctic ice sheet to climate. However, it is widely recognised that the age model for Antarctic melt observations has more than 10-20% uncertainties associated with the complications of radiocarbon dating, including the age of marine reservoirs (e.g. 1000 year age differences or so) or exposure ages (e.g. 10-20% of age differences due to production rate estimates and other factors). Thus, their conclusion regarding the timing of the Antarctic melt is not convincing because it was derived without considering these factors.

We respectfully disagree that the age model for Antarctic melt observations has 10-20% uncertainties because of radiocarbon dating associated with the age of the Antarctic marine reservoir correction or more generally with the global marine reservoir correction. No RSL data south of 60°S are used in this study. We use only low- to mid-latitude RSL data. The average of all radiocarbon age uncertainties used in this study is 254 years, but age uncertainties are taken into account via the WRSS calculation that we describe in the methods. We note that the age model for the West Antarctic temperature curves to which we compare our posterior Antarctic ice volume are constructed by counting ice layers back to 31 thousand years and are not radiocarbon dated for any temperatures during the Holocene. We also note that the ensemble of thermomechanical PISM Antarctic Ice Sheet simulations we employ (Albrecht et al. 2020, 2021) are not data constrained and therefore not subject to radiocarbon age uncertainties. Additionally, our cross-correlational analysis, which determined the amount of lag between West Antarctica surface air temperature records and Antarctic ice volume, included tests for statistical significance. Finally, not all sea-level data that are radiocarbon dated are affected by the marine reservoir correction: most terrestrial limiting, and any index points with dates from terrestrial material, do not require a marine reservoir correction. We therefore believe that our comparison of the timing of Antarctic ice volume to West Antarctic temperature is robust.

Instead of taking a "big data approach", I strongly recommend the authors to show region by region comparisons between the currently proposed GMSL-based predicted sea level and observations. This will be a very important contribution to the community, but it will require much more space offered by Nature Communications.

We thank Reviewer 3 for this suggestion. While our paper as written did not include a method to reconstruct relative sea level on a region-by-region level, our algorithm is extensible to such a procedure, though doing so required that we rerun all simulations, which led to the multi-month period for this rebuttal. We have added text in the Methods to describe our approach (lines 489 and following):

We next demonstrate our algorithm’s efficacy at producing RSL curves that match observations within uncertainties. A selection ($n=20$) of sites s of size 5 degrees lat/lon are chosen in regions of high data density, and for each site and ice model the RSL curve RSL_i which best fits nearby data, i.e. the curve with the minimum χ^2_{is} , is selected. These curves are merged with the χ^2_i weights assigned to the curves’ associated ice models in order to produce a posterior RSL curve RSL_s :

$$RSL_s = \sum_{i=1}^I RSL_i * (1/\chi^2_i) / \sum_{i=1}^I (1/\chi^2_i) \quad (9)$$

The procedure balances local information --- included through the selection of RSL_i curves via χ^2_{is} minimization --- with the global information contained in the χ^2_i weights. Data-model comparisons are plotted in Supplemental Figs. S2-S5 for each site. We don’t include data from the entire 5 degree lat/lon box since GIA varies across this scale making it impossible to show a single posterior RSL curve for each site. Instead, we only plot RSL data within 1 degree latitude/longitude of the center of each site and plot modeled RSL for this location.

Comparisons between RSL data and modeled RSL support the aggregated misfit calculations (see Methods Section 4 and Extended Fig. 9). For instance, higher misfit scores in Southeast Asia correspond to systematic misfit trends in Fig. S3F (Vietnam) and S4E (Shanghai, China) and align with the lack of correction for local subsidence in these data (e.g. Zong et al. 2004). Beyond Figs. S4E/ S3F, modeled RSL curves fit virtually all RSL data within uncertainties save in Fig. S2E/F (New Jersey/North Carolina). There RSL data consistently plot above modeled RSL in the early-mid Holocene, a trend that mirrors the offset between GIA models and RSL data prior to 4 ka noted by Walker et al. (2022), which they attribute to inaccuracies in Northern Hemisphere ice histories—an assessment we support.

For instance, higher misfit scores in Southeast Asia correspond to systematic misfit trends in Supplemental Figs. S4F (Vietnam) and S4E (Hainan and Shanghai, China) and align with the lack of correction for local subsidence in these data (e.g. Zong et al. 2004). Similar misfits due to lack of subsidence correction of deltaic index points occur in Supplemental Fig. S3E (c.f. Marriner 2012). All misfitting index points from these regions that plot outside the credible interval of our RSL reconstruction are from papers not standardized following the HOLSEA protocols. This misfit pattern motivates future work to standardize RSL data in these regions. Beyond Supplemental Figs. S4E/F and 3E, modeled RSL curves fit virtually all RSL data within uncertainties save in Fig. S2E/F (New Jersey/North Carolina). There RSL data consistently plot above modeled RSL in the early-mid Holocene, a trend that mirrors the offset between GIA models and RSL

data prior to 4~ka noted by ref.\citep{walker5000yearRecordRelative2022}, which they attribute to imprecisions in ice histories---an assessment we support.

Minor points:

1) Some minor points are that the figures do not clearly show the magnitude of the higher mid-Holocene GMSL due to the tick marks and the size of the figures. The plots should be enlarged to show the differences between the GMSL models, especially in the mid to late Holocene. The plot also ignores significant figures as the authors argue that their GMSL can resolve less than 0.5m sea level.

We thank Reviewer 3 for this recommendation and agree that figures should be legible. We have extended the inset plot in Figure 2 to 6 ka and the range to -1 to +1 m. In response to Reviewer 3's concern about the ability to distinguish sub-half meter scale changes from our plots, we have also added a higher-resolution grid in the inset plots.

2) The thermosteric sea level correction cannot be made precisely, as we also have large uncertainties in the Holocene sea surface temperature using climate models (cf. CMIP). Therefore, the associated uncertainties are also likely to be underestimated.

We thank Reviewer 3 for their attention to this point. A key feature of our thermosteric sea-level reconstruction is that it does not use sea-surface temperatures. Rather, we use new reconstructions of mean ocean temperature derived from ratios between krypton, xenon, and nitrogen (see Shackleton et al. 2021 for details). Shackleton and colleagues quantify uncertainties associated with their method in their data, which we propagate into our estimates. As is visible in Extended Figure 7, our median thermosteric sea-level estimate hovers within 10 cm of present levels throughout the Holocene but with large uncertainties. The contribution is therefore small, but we disagree that the uncertainties are underestimated.

I have been asked to evaluate the authors' responses to Rev. 3, especially the reviewer's concern that the study's newly proposed GMSL can't uniquely eliminate the "apparent overshoot" of GMSL. (It took me a while to understand what this was referring to since the authors don't use this terminology, but it's basically saying that late Holocene sea level did not exceed 1 m).

The authors do not address this specific question. Instead, they point to the various novelties of the paper, thus addressing the reviewer's earlier comment implying that the only novelty of the paper is the inclusion of new data, and they are correct in doing so. However, regarding the "apparent overshoot" of GMSL, the authors only respond with "We agree with Reviewer 3 that our results don't uniquely constrain GMSL to have been higher than 1m during the Holocene" (although they must mean lower than 1 m, not higher). This is not clearly stated in the paper, yet a key take-home point of the paper (Section 4) is to use this constraint to conclude that "Future GMSL will more likely than not ($P > 0.5$) exceed maximum Holocene GMSL by 2060 under all emissions scenarios (Fig. 4A). By 2150, future GMSL will likely ($P > 0.66$) be higher than peak Holocene GMSL under low emissions (SSP1-2.6) and very likely ($P > 0.9$) higher under high emissions (SSP5-8.5)." I agree with the authors that the strength of the paper is a new and robust estimate of Holocene sea level that in places differs from previous reconstructions, which is to be expected given the improvement in the models and the large increase in data, and in this regard I support publication. However, I think the authors still need to explicitly address Rev. 3's comment about the "overshoot," especially in the context of Section 4.

We thank Reviewer 4 for this fair assessment of our work. We interpret Reviewer 3's comment,

"I am not convinced that the approaches can solve the fundamental issue behind the topic, namely the newly proposed GMSL can uniquely eliminate the "apparent overshoot" of GMSL as small as less than 1 m."

to be a critique of our model for its inability to constrain Holocene GMSL to have been higher than present. To slightly rephrase the comment with less opaque syntax:

"I am not convinced that the approaches can solve the fundamental issue behind the topic, namely **that** the newly proposed GMSL can uniquely eliminate **an** "apparent overshoot" of GMSL **that is** as small as less than 1 m."

To directly respond to this comment, we have added the following sentence to the main text (Line 108):

While our results favor GMSL overshooting pre-industrial levels by up to 1 meter in the mid-late Holocene, they also leave open the possibility ($P = 0.25$) that Holocene GMSL did not exceed pre-industrial levels.